# Sturnidae *sensu lato* Mitogenomics: Novel Insights into Codon Aversion, Selection, and Phylogeny

**DOI:** 10.3390/ani14192777

**Published:** 2024-09-26

**Authors:** Shiyun Han, Hengwu Ding, Hui Peng, Chenwei Dai, Sijia Zhang, Jianke Yang, Jinming Gao, Xianzhao Kan

**Affiliations:** 1Anhui Provincial Key Laboratory of the Conservation and Exploitation of Biological Resources, College of Life Sciences, Anhui Normal University, Wuhu 241000, China; hansy@ahnu.edu.cn (S.H.); dinghengwu@caas.cn (H.D.); zhangsijia@ahnu.edu.cn (S.Z.); ajiankebc@wnmc.edu.cn (J.Y.); gaojming@ahnu.edu.cn (J.G.); 2Teaching and Research Office of Evidence-Based Medicine, Wannan Medical College, Wuhu 241002, China; 3Anhui Academy of Medical Sciences, Anhui Medical College, Hefei 230061, China; daicw@mail.ustc.edu.cn; 4School of Basic Medical Sciences, Wannan Medical College, Wuhu 241002, China; 5The Institute of Bioinformatics, College of Life Sciences, Anhui Normal University, Wuhu 241000, China

**Keywords:** mitogenome, starlings, Mimidae, Buphagidae, codon usage, codon aversion motifs, phylogenetic inferences

## Abstract

**Simple Summary:**

Sturnidae *sensu lato* (Muscicapoidea) consists of Sturnidae, Mimidae, and Buphagidae. However, the deep-node evolutionary relationships within this group remain undetermined, and few efforts have been made to elucidate their mitogenomic evolution. Here, we focus on Sturnidae *sensu lato* and present five newly sequenced mitogenomes. Together with analysis of publicly available data, comprehensive analyses of mitogenome features, codon usage and aversion, RNA and CR structures, and phylogeny are performed. In conclusion, we determine the basic organizations of Sturnidae *sensu lato* mitogenomes, demonstrate the pervasiveness of natural selection in forming the CUB patterns, depict the RNA secondary structures, and construct both the backbone and deep-node relationships within Sturnidae *sensu lato*. The main finding of our research is obtained through codon aversion motifs (CAM) analyses. The surprisingly unique CAMs from 11 mt PCGs for each species offer new opportunities for the identification of the molecular species identification of these taxa. This work can shed new light on the mitogenomic evolution of Sturnidae *sensu lato*.

**Abstract:**

The Sturnidae family comprises 123 recognized species in 35 genera. The taxa Mimidae and Buphagidae were formerly treated as subfamilies within Sturnidae. The phylogenetic relationships among the Sturnidae and related taxa (Sturnidae *sensu lato*) remain unresolved due to high rates of morphological change and concomitant morphological homoplasy. This study presents five new mitogenomes of Sturnidae *sensu lato* and comprehensive mitogenomic analyses. The investigated mitogenomes exhibit an identical gene composition of 37 genes—including 13 protein-coding genes (PCGs), 2 rRNA genes, and 22 tRNA genes—and one control region (CR). The most important finding of this study is drawn from CAM analyses. The surprisingly unique motifs for each species provide a new direction for the molecular species identification of avian. Furthermore, the pervasiveness of the natural selection of PCGs is found in all examined species when analyzing their nucleotide composition and codon usage. We also determine the structures of mt-tRNA, mt-rRNA, and CR structures of Sturnidae *sensu lato*. Lastly, our phylogenetic analyses not only well support the monophyly of Sturnidae, Mimidae, and Buphagidae, but also define nine stable subclades. Taken together, our findings will enable the further elucidation of the evolutionary relationships within Sturnidae *sensu lato*.

## 1. Introduction

As part of the Muscicapoidea superfamily of passerine birds, the Sturnidae *sensu lato* consists of starlings (Sturnidae), mimids (Mimidae), and oxpeckers (Buphagidae) [1]. Within this group, starlings are most closely related to mimids, and these two lineages form a sister relationship with oxpeckers. The Old World family Sturnidae contains approximately 123 species in 35 genera, with centers of diversity in Africa and Southeast Asia [2,3,4]. In contrast, Mimidae consists of New World species (34 species in 10 genera) that cover large parts of the West Indies and North, Central, and South America [2,4]. Furthermore, Buphagidae, comprising only two *Buphagus* species, is endemic to the savannah of Sub-Saharan Africa [5]. Mimidae and Buphagidae, formerly treated as subfamilies within Sturnidae, are now usually considered to be two distinct families [3,4,6]. Over the years, considerable research has been undertaken to address the evolutionary history of Sturnidae *sensu lato* [1,2,7,8,9,10]. Unfortunately, owing to extreme morphological variations and a high level of morphological homoplasy, the deep-node phylogenetic relationships within this group remain poorly understood [2].

Mitochondria play a vital role in various biological processes, such as energy production, calcium homeostasis, signaling, apoptosis, and cellular proliferation [11,12,13,14]. In all animal lineages (expect only for the recently discovered *Henneguya salminicola* [15]), mitochondria possess their own genomes and independent transcriptional and translational machineries. In vertebrates, the mitogenome is a circular double-stranded molecule with a relatively stable gene organization, containing 37 coding genes (two for rRNA, 22 for tRNA, and 13 for PCGs) and one noncoding region (CR) [16,17]. Significantly, due to their low recombination and rapid evolution rates, mitogenomes have been extensively utilized as molecular markers for evolutionary research [18,19,20,21,22,23]. In recent years, the significant development of next-generation sequencing technologies has led to the attainment of an increasing amount of sequencing data for avian mitogenomes. So far, a total of 1608 mitogenome sequences from birds have been published in the National Center for Biotechnology Information (NCBI) database, with sizes ranging from 15,523 bp (*Coturnicops noveboracensis,* Gruiformes) to 25,624 bp (*Nettapus auritus*, Anseriformes) and a mean size of 17,121 bp. Coupled with such a burgeoning availability of data, the field of avian mitochondrial phylogenomics (mitophylogenomics) has undergone unprecedented growth [24,25,26,27,28,29,30,31,32,33,34,35,36,37]. However, the mitogenomes for only five of the 159 known Sturnidae *sensu lato* species have been reported. Therefore, to gain further insights into the characteristics of the mitogenomes of these closely related taxa and their phylogenetic implications, more samples are needed.

In this paper, we report five new complete mitogenome sequences of Sturnidae *sensu lato* taxa, as well as two low-coverage nuclear genomes of *Spodiopsar* (Sturnidae) species. Together with the sequences available in the NCBI database, these new sequences are utilized for further analyses. Based on both nuclear and mitochondrial sequence data, we attempt to address the following: (1) the nucleotide features and organization of Sturnidae *sensu lato* mitogenomes, (2) the codon usage bias and aversion patterns of mitochondrial genes, (3) the structures of mitochondrial tRNA and rRNA genes, and (4) the phylogenetic relationships within and among the three related families (Sturnidae, Mimidae, and Buphagidae). As far as we know, the current work is the first to explore the mitogenome-wide evolution of these taxa.

## 2. Materials and Methods

### 2.1. Sampling, DNA Extraction, Sequencing, and Assembly

This study displayed five new mitogenome sequences. Four of the five sequences were obtained from the frozen muscle tissues of *Acridotheres cristatellus*, *A. tristis*, *Gracupica nigricollis*, and *Spodiopsar cineraceus*. These samples were collected from the Ningguo Museum of Natural History (NMNH), Xuancheng, China. The remaining sequence (*Mimus polyglottos*) was obtained from the NCBI database using third-party annotation (TPA).

Whole-genomic DNA was extracted from muscle using phenol–chloroform using the standard protocol [38]. New PCR primers (Appendix A) for amplification and sequencing were designed based on the sequences available in GenBank. The entire mitogenomes were amplified into long overlapping segments using the LA PCRTM Kit (Takara, Dalian, China). The resulting amplicons were used as templates for nested PCR with specific primer sets. After being purified, the PCR products were sequenced on the ABI-PRISM 3730xl platform. The resulting sequences were assembled in Sequencer v4.14 (Gene Codes Corp., Ann Arbor, MI, USA). Furthermore, the short-paired reads files for *Mimus polyglottos* were derived from the NCBI SRA database. Subsequently, the mitochondrial reads were extracted and de novo assembled using the GetOrganelle v1.7.5 [39,40]. The resulting filtered circular *de Brujin* mitogenome graphs were checked using Bandage v0.8.1 [41].

The current study additionally presents low-coverage nuclear genomic data from two species of *Spodiopsar* (*S. cineraceus* and *S. sericeus*) for further phylogenetic study. Frozen muscle tissues for these two species were also provided by NMNH. The genomic DNA extraction was conducted using the E.Z.N.A. Tissue DNA kit (Omega Bio-Tek, Norcross, GA, USA). The library was generated using the TruSeq DNA PCR-Free Library Prep Kit (Illumina, San Diego, CA, USA), which was then sequenced using Illumina Hiseq X Ten (Illumina, San Diego, CA, USA) under the 150-paired-ends strategy with 350 bp insert sizes.

### 2.2. Gene Annotation and Comprehensive Sequence Analyses

The mitogenomes were annotated with GeSeq [42]. The mitochondrial genes were named according to the HUGO Nomenclature Committee (HGNC) [43]. The species scientific names followed the Clements Checklist 2019 [4]. The mitogenomic map of Sturnidae *sensu lato* was depicted using the CGview comparison tool (https://paulstothard.github.io/cgview_comparison_tool/, accessed on 24 August 2024) [44]. The nucleotide composition was calculated using MEGA X [45]. The GC content of the synonymous third codon positions (GC3s) and the nucleotide polymorphisms for each species were determined with DnaSP v6.12.03 [46]. PAML v4.9 (F3X4 and M0 models) was used to assess the nonsynonymous substitution rate (dN), synonymous substitution rate (dS), and dN/dS. Specifically, the value of dN/dS quantified the selection pressure, with <1, =1, and >1 indicating purifying, neutral, and positive selection, respectively. The relative synonymous codon usage (RSCU) values and the effective number of codons (ENCs) were calculated using CodonW v1.4.4 [47]. Subsequently, the aversive codons were determined according to the rule that codons with RSCU = 0. The ENC-GC3 plot, principal component analysis (PCA), and the parity rule 2 (PR2) plot were plotted using R x64 4.0.2. The tRNA secondary structures were created using the tRNAscan-SE v2.0.3 [48], and the rRNA secondary structures were predicted based on those from other birds [25,49]; the numbering of the helixes of rRNAs followed the method proposed by Cannone et al. [50].

### 2.3. Phylogenetic Inferences

To assess the evolutionary affinities among Sturnidae *sensu lato*, two datasets were established. One dataset comprised 13 mitogenomic PCGs from 10 available Sturnidae *sensu lato* species. Meanwhile, the other was generated by combining the nuclear and mitochondrial data of 132 Sturnidae *sensu lato* species from 45 genera, including five mtDNA loci (MT-ATP6, MT-ATP8, MT-CO1, MT-CO2, and MT-ND2) and eight nuclear loci (Fib5, Fib7, RDP1, TGFB2-4, ODC, GAPDH-11, myo, and RAG-1) (Appendix A). Furthermore, two species from the family Muscicapidae, namely, *Ficedula albicollis*, and *F. zanthopygia*, were employed as outgroups for both datasets. Additionally, to obtain the optimal resolution and clade confidence, three frequently used methods were employed to obtain a combined phylogenetic inference, i.e., maximum likelihood (ML), maximum parsimony (MP), and Bayesian inference (BI).

The ML analyses were conducted using RAxML 8.2.12 [51], with 100 random starting trees, 1000 bootstrap replicates (under GTRCAT model), and a bootstrap convergence criterion. PAUP* version 4.0a168 was used to perform the MP analyses [52] with heuristic search settings: 10 random-stepwise-addition replicates with tree-bisection reconnection (TBR) branch swapping, and 1000 bootstrap replicates. Subsequently, prior to the BI analyses, the model best suited to each genomic locus was determined using ModelTest-NG 0.1.6 [53]. For the BI inference, two simultaneous runs and four independent Markov chains were run for 10,000,000 generations (sampling every 1000th generations) using MrBayes 3.2.7a [54]. Lastly, the over-100 effective sample sizes (ESSs) of all parameters were considered to check the convergence of chains.

## 3. Results

### 3.1. NGS Data Information and Mitogenome Organization

In the present study, we obtained five new mitogenomes of Sturnidae *sensu lato*, and their GenBank accession numbers can be found in Appendix A. All five of our newly generated mitogenomes were supercoiled, double-stranded, and circular macromolecules. Comparatively, we analyzed all ten mitogenomes that were obtained from Sturnidae *sensu lato*. It should be noted that the publicly available mitogenome sequence of *Gracupica nigricollis* (JQ003192) was not included here, as these data were actually derived from both *Gracupica nigricollis* and *Acridotheres cristatellus* [55]. The results showed that the sizes of the mitogenomes ranged from 16,780 bp (*Toxostoma redivivum*) to 16,845 bp (*Leucopsar rothschildi*) (Table 1). Similar to most passerine birds, the nucleotide compositions of their mitogenomes (heavy strand) were slightly biased toward A and T, while the total of AT content ranged from 51.85% (*Buphagus erythrorynchus*) to 52.95% (*Gracupica nigricollis*) (Table 1). As expected, most of these genes were encoded on the heavy strand except for eight tRNA genes and MT-ND6 (Figure 1a). Including stop codons, the total length of the PCGs of each mitogenome was identical (11,400 bp), with the AT content ranging from 50.65% to 52.13% (Table 1).

In addition, our study also provided new NGS data for two *Spodiopsar* species. A total of 35,782,311 and 33,050,441 clean reads were sequenced from the *S. cineraceus* and *S. sericeus* libraries, respectively. All clean reads that were yielded have been submitted to the NCBI SRA database under the accession numbers SRR10053857 (*S. cineraceus*) and SRR10053850 (*S. sericeus*). Here, we obtained eight new nuclear loci for *S. cineraceus* and *S. sericeus* from the NGS data.

### 3.2. Codon Usage Bias (CUB) and Codon Aversion of Mitochondria PCGs

To elucidate the codon usage patterns of the mitogenomes among Sturnidae *sensu lato* taxa, we assessed the RSCU values, PR2 plots, PCA, and ENC-GC3s plots for the ten involved taxa.

With the exclusion of initial and stop codons, the RSCU of the overall PCGs was compared among the Sturnidae *sensu lato* mitochondria. Displaying a relatively high level of diversity, the RSCU values ranged from 0.04 (codon ACG of *Spodiopsar cineraceu*) to 4.17 (codon CGA of *Toxostoma redivivum*) (Figure 2a). Notably, across the synonymous codons, those ending with A or T were mostly favored.

For each of the 13 mitochondrial PCGs, the PR2 plots were further produced by restricting four-fold degenerate codon families to third-codon sites (Figure 2b). For the overwhelming majority of the mt PCGs (12 of the 13), the points fell into the quadrant II and were far off the central axes, implying a strong AC bias at the third-codon positions of these codons. Most remarkably, MT-ND6 showed the exact opposite pattern in quadrant IV, demonstrating a TG-ending bias.

Surprisingly, the codon aversion analyses revealed the most striking interspecies disparities within Sturnidae *sensu lato*. For all 11 mt PCGs with a size over 300 bp, the codon aversion motifs differed among the ten species examined (Figure 3a). Meanwhile, it was interesting that the numbers and corresponding amino acids of the aversion codons were identical among the three families (Figure 3b). Within this pattern, Sturnidae seemed to share a higher level of similarity with Mimidae than Buphagidae.

The ENC analyses were performed to further explore the codon usage bias. The ENC values of the ten species were found to possess a narrow range of 36.88 to 42.02 (Appendix A). Collectively, the results revealed no strong codon usage bias within Sturnidae *sensu lato*. Moreover, PCA was employed to determine the variety of codon usage patterns based on ENC (Figure 2c). PC1 explained 42.7% of the variance, while PC2 accounted for 18.1%. As shown in Figure 2d, the points of the ENC-GC3s plot dipped well below the curve, demonstrating that the codon usage pattern of mitogenomes within Sturnidae *sensu lato* might result from natural selection.

### 3.3. Secondary Structure of tRNAs, rRNAs, and CRs

All ten mitogenomes contained 22 tRNA genes with a total length ranging from 1540 bp (*M. polyglottos* and *S. vulgaris*) to 1546 bp (*B. erythrorynchus*) (Figure 4 and Table 1). In addition, all the tRNAs could be folded into typical cloverleaf secondary structures, except for MT-TS2 (*trnS*-AGY), which lacks the stem of the dihydrouridine (DHU) arm. Moreover, three unpaired nucleotides (A, C, and C) were observed in the TΨC stem of MT-TF within Sturnidae *sensu lato* mitogenomes (Figure 4).

Similar to the typical avian mitogenomes, MT-RNR1 and MT-RNR2 were separated by MT-TV. The size and AT contents of MT-RNR1 and MT-RNR2 are presented in Table 1. Here, we used *A. cristatellus* as an example to predict the secondary structures of MT-RNR1 and MT-RNR2, which comprised 49 helices (three domains) and 71 helices (six domains), respectively (Figure 5 and Figure 6). Further comparative analysis showed that the overall secondary structures of MT-RNR1 and MT-RNR2 were conserved well among the birds, except for a few small differences. Among our sampled species, the CRs were located between MT-TT and MT-TF, with their size ranging from 1208 bp (*T. redivivum*) to 1254 bp (*L. rothschildi*) (Table 1); this also contained three conserved domains and eight conserved blocks, namely, boxes F, E, D, C, CSBa, and CSBb, b, and B in Domain II (Figure 7).

### 3.4. Rates and Patterns of Mitochondrial Gene Evolution

To examine the evolutionary patterns of the mitogenomes within Sturnidae *sensu lato* further, we calculated the variable sites, nucleotide diversity (π), dN/dS, and ts/tv of 13 mt PCGs. The results showed that MT-ND2 had the most variable sites (37.66%), followed by MT-ATP8 (34.52%), MT-ND6 (34.30%), and MT-ND1 (33.74%) (Table 2). In contrast, the cytochrome c oxidase genes and cytochrome b gene (MT-CO1, MT-CO2, MT-CO3, and MT-CYB) had lower percentages among the PCGs (Var. sites [%] = 23.79, 28.65, 27.35, and 25.55, respectively). Furthermore, the π values range from 0.09015 (MT-CO1) to 0.13797 (MT-ND2), with the ts/tv ratios varying from MT-ATP8 (5.4548) to MT-ND4 (12.7131) (Table 2).

Our analyses revealed a dN/dS pattern with a clear signature of purifying selection on the mitogenomes within Sturnidae *sensu lato* (Table 2). Compared with other PCGs, MT-ATP8 had the highest dN/dS ratio (0.25631); it also had some highly variable sites with extreme changes in the properties of amino acids. In contrast, the lowest dN/dS ratio was estimated for MT-CO1 (0.00579).

### 3.5. Phylogenetic Implications

The phylogenetic affinities among the Sturnidae *sensu lato* taxa were inferred using two datasets (mitogenomic dataset and multilocus dataset). The best-fit models were also assessed (Table 3 and Appendix A). In addition, the substitution saturation test demonstrated that all three codon positions were unsaturated (Table 4 and Appendix A).

Firstly, according to the 12-mitogenome dataset (ten Sturnidae *sensu lato* taxa and two outgroups), the phylogenetic topologies of the ML, MP, and Bayesian trees were nearly identical (Figure 1b). The monophyly of Sturnidae *sensu lato* was also confirmed (100% in ML and MP, and 1.00 in BI). The family Buphagidae formed a sister-group with (Sturnidae + Mimidae). Notably, the clade of two species of *Spodiopsar* (*S. cineraceus* and *S. sericeus*) was only weakly supported (55% in ML, 60% in MP, and 0.94 in BI).

Secondly, to obtain more densely sampled phylogenies of Sturnidae *sensu lato*, a combined dataset (five mitochondrial genes and eight nuclear genes) was generated; this represented 83% of the known species and almost all currently recognized constituent genera. As shown in Figure 8, the trees that resulted from the three methods exhibited similar topologies. For deeper-node level, the Sturnidae family could be further classified into six major subclades: (1) *Phillipine Rhabdornis*, (2) South Asian/Pacific Starlings, (3) Eurasian Starlings, (4) Red-winged Starlings, (5) African Starlings, and (6) Amethyst and Madagascar Starlings (Figure 8). It was proven that all the former five had monophyly, with strong support; the only exception was African Starlings, which had moderate support (84% in MP, 1.00 in BI, and <50 in ML). Furthermore, the monophyly of Amethyst and Madagascar Starlings was not found in either analysis ((BS_ML_) = 55, (BS_MP_) = 73, (PP) = 0.96).

Within the African Starlings subclade, the monophyly of *Lamprotornis*, *Poeoptera*, and *Onychognathus* was strongly supported by the ML and BI methods (Figure 8). Furthermore, *Pholia sharpii* is currently the only species of the monotypic genus *Pholia* in both Clements Checklist v. 2019 [4] and Birds of the World v. 1.0 [59]. However, it was embedded within the genus *Poeoptera*, forming a well-supported clade.

Furthermore, the results showed that the subclade Eurasian Starlings comprised six polytypic genera (*Acridotheres*, *Agropsar*, *Gracupica*, *Spoliopsar*, *Sturnia*, and *Sturnus*) and five monotypic genera (*Creatophora*, *Fregilupus*, *Leucopsar*, *Pastor*, and *Sturnornis*). The monophyly of the six polytypic genera was well supported by all estimators (≥98, ≥80, and 1.00 in ML, MP, and BI, respectively). In addition, *A. tristis* and *A. ginginiianus* are recognized as a sister group to the remaining taxa of *Acridotheres*.

The South Asian/Pacific Starlings subclade consisted of 10 genera, five of which were monotypic (*Ampeliceps*, *Sarcops*, *Goodfellowia*, *Enodes,* and *Scissirosrum*). According to the IOC World Bird List v. 11.1 [60], Apo Myna (*G. miranda*), which is endemic to the Philippines, belongs to the genus *Basilornis*, namely, *B. mirandus*. Nevertheless, *B. mirandus* and *B. celebensis* did not form a monophyletic group. On the contrary, the monophyly of *G. miranda* and (*S. albicollis* + *S. calvus*) was strongly supported (Figure 8).

For the family Mimidae, all 34 recognized species were included in the present analyses. In our trees, this family could be divided into two well-supported monophyletic subclades: one comprising Mockingbirds and continental Thrashers and the other containing Catbirds and Caribbean Thrashers. In the former subclade, the clade of *M. graysoni*, *M. gilvus* and *M. polyglottos* was recovered with strong support ((BS_ML_) = 100, (BS_MP_) = 99, (PP) = 1.00). Furthermore, the latter subclade comprised five monotypic genera (*Allenia*, *Dumatella*, *Margarops*, *Melanoptila*, and *Ramphocinclus*) and two bitypic genera (*Cinclocerthia* and *Melanotis*). However, the relationships among *Dumetella*, *Melanoptila*, and *Ramphocinclus* were not recovered in any of our trees.

## 4. Discussion

The present study reported five mitogenomes of Sturnidae *sensu lato* and two low-coverage nuclear genomes of *Spodiopsar* (Sturnidae) species for the first time. Then, multidimensional analyses were performed by combining all the available mitogenomic data of Sturnidae *sensu lato*, including general mitogenomic characteristics, codon usage and aversion patterns, RNA structures, substitution rates, and phylogenetic tree constructions.

Codon usage bias (CUB), a gene- and taxon-specific phenomenon, has long been considered essential in understanding the evolution of both genes and taxa [23,61,62]. The ENC is often used as a measure of the bias caused by equal codon usage in a gene [63]. Generally, the range of ENC values is 20 to 61, signifying extreme codon usage bias to no bias at all [64]. It is worth mentioning that genes with ENC ≤ 35 are considered strongly biased [65,66]. Here, we showed that the mt genes of the ten investigated Sturnidae *sensu lato* taxa have no strong codon usage bias. This finding is actually similar to the result of a previous study focusing on the mitogenomes of Odonata [67], but is different from the results of a study on Lepidoptera species that presented obvious bias (ENC ranged from 27.7 to 33.8) [68]. Unfortunately, there is currently very little information regarding this issue available. Thus, further large-scale investigations of the mitogenomic ENC in birds and other animals are needed. In recent years, analyzing the ENC versus GC3s plot analysis has proven to be a highly efficient tool for verifying the main driving factor of CUB (natural selection or mutational bias) [63,64,65,66,69]. If mutational bias is the single factor, the analyzed points will lie on or just below the ENC curve. Alternatively, if natural selection is the single factor, the points will fall below the expected curve [63,64,65,66,69]. For Sturnidae *sensu lato* mitogenomes, natural selection was presumed to be the main cause of their CUB patterns. Notably, such conclusions have also been reported in the mitogenomes of other animals, e.g., odonates [67] and ladybirds [65]. As we know, codon usage may vary significantly between genomes, even between genes within the same genome. Thus, to explore the impact of codon usage patterns on avian mitogenomes, more complicated analyses are required.

One novel phylogenetic characteristic and potential molecular marker, the codon aversion motif (CAM), was recently proposed by Miller et al. [70]. Then, these strong evolutionary implications were further proven across the tree of life [71]. More importantly, our previous work on the codon aversion of plastomic genes in various taxa subsequently discovered abundant taxon-unique molecular markers, including *Aeonium* [72], *Monanthes* [72], and *Crassula* [73] (Crassulaceae), as well as *Bletilla* [74] (Orchidaceae). In this work, we first focused on the mt PCGs of Sturnidae *sensu lato*. Remarkably, according to our results, the interspecific disparities in avian mt genes were even much stronger than in plant cp (chloroplast) genes. Most surprisingly, for all examined 11 mt PCGs, the ten species investigated possessed completely different CAMs. All of these identified CAMs have the potential to serve as unique markers for corresponding Sturnidae *sensu lato* species. Meanwhile, this observation might indicate the potential use of mt-gene CAM in the molecular species identification and evolutionary demonstration.

As is well known to us, mitochondria have their own set of mtDNA that is distinct from nDNA. Moreover, they embody the mt-rRNAs and mt-tRNAs necessary for synthesizing essential proteins [75]. As previously noted, the structural features of tRNA provide important insights into their biological function [76]. The present research presumed that all the mt-tRNAs fold as typical cloverleaf secondary structures, with the exception of known MT-TS2 (*trnS*-AGY), which lacks the stem of the dihydrouridine (DHU) arm. As documented, this unusual D-arm-lacking tRNA exists in almost all metazoan mitochondria, and has been experimentally verified to be functional in translation [77,78,79,80,81,82,83], despite having a much lower translational ability compared with MT-TS1 (*trnS*-UCN) [77,78]. Furthermore, we also detected unpaired nucleotides in the TΨC stem of MT-TF. It is noteworthy that, for Aves, a total of five matching patterns in the TΨC stem of MT-TF had been discovered by previous studies [84,85,86]. These patterns were presumed to possess evolutionary significance [86]. Most importantly, differing from the C-insertion pattern (form b that displayed in Figure 4 of the study by Ma et al. [86]) of many Passerine birds, the MT-TF examined in our study featured an AC-C mismatch. We presume that this novel matching pattern might be a derivative of the C-C mismatching pattern (form e displayed in Figure 4 of the study by Ma et al. [86]). Collectively, these discussed findings suggest that the matching patterns of MT-TF might be a potential taxonomic marker for birds.

Numerous efforts have hitherto been dedicated to interpreting the relationships within Sturnidae *sensu lato* [2,3,9,10]. Nevertheless, a complete picture of its phylogeny remains lacking, and a denser sampling strategy and multidimensional datasets are required. In this study, the topologies of the trees obtained using the three methods were identical, except for the positions of only two species (*Lamprotornis iris* and *Aplonis grandis*), which were both weakly supported. Indeed, such high similarity seems to be reasonable. The assessment of Torres et al. [87] indicated that the incongruences between the trees created using different methods might more frequently occur in higher-level phylogenies (e.g., phylum/kingdom).

A clear and stable backbone of Sturnidae *sensu lato* was disentangled by our mt-nuclear combined phylogeny; namely, it was found that Buphagidae is a sister to (Sturnidae + Mimidae). This result is congruent with the conclusions of Lovette and Rubenstein [2]. More importantly, the previously undetermined relationships among four subclades (African Starlings, Red-winged Starlings, Amethyst and Madagascar Starlings, and Eurasian Starlings) were established [2]. Here, our Bayesian tree illustrated a closer relationship between African and Red-winged Starlings ((PP) = 1.00), which together formed a sister group with the Amethyst and Madagascar Starlings subclade ((PP) = 0.91). The Eurasian Starlings subclade was a sister to these three subclades.

Our phylogenetic inference also advanced our understanding of some deep-node affinities compared with previous work. For instance, within the African Starlings subclade, three species (*Lamprotornis albicapillus*, *L. bicolor*, and *L. fischeri*) that were previously treated as members of the genus *Spreo* [2,3] were found to be nested within the remaining *Lamprotornis* birds in our trees. Moreover, we gained insights regarding the Eurasian Starlings subclade. Differing from the mitochondrial tree produced by Zuccon et al. [10], our results clearly demonstrated that *Acridotheres javanicus* is a sister to *A. fuscus*, not to *A. cinereus*.

Beyond the progress that has been made, a few topological uncertainties persist. For instance, the phylogenetic relationships within the Catbirds and Caribbean Thrashers subclade remain controversial. For example, three previous studies suggested a sister relationship between *Dumetella* and *Ramphocinclus* [2,8,88]. In addition, the sister relationship between the genus *Dumetella* and the genus *Melanoptila* was strongly supported by a recent report based on nuclear genomic data [7]. Unfortunately, neither of the two sister relationships were found in any of our trees. To resolve this uncertainty, more data from this clade are needed in the future.

## 5. Conclusions

This study provides comprehensive insights into the mitogenomic diversities and evolutionary relationships among Sturnidae *sensu lato*. Based on our five newly reported mitogenomes and all currently publicly available data, we conducted comparative analyses and phylogenetic inferences. The general organization and gene content of the examined mitogenomes were clarified. Additionally, we depicted the overall CUB patterns of the mt PCGs, including the A/T-ending codon preference (via RSCU analyses), the major AC bias for the third positions of four-fold degenerate codons (using PR2 plots), and the pervasiveness of natural selection over mutational bias (using ENC-GC3s plots). These insights might enlighten us regarding the evolution of Sturnidae *sensu lato* to a degree. The key finding of this study was drawn from CAM analyses. The unique motifs found in each species provide new opportunities for the molecular species identification of avian. For structural analyses, except for MT-TS2, all tRNAs were presumed to possess classic cloverleaf structures. MT-RNR1 and MT-RNR2 consisted of three domains with 49 helices and six domains with 71 helices, respectively. The CRs in all investigated species consisted of three conserved domains. Furthermore, our phylogenetic reconstructions not only supported the monophyly of Sturnidae, Mimidae, and Buphagidae, but also revealed nine subclades among Sturnidae and its related taxa, with high support values. In summary, the conclusions presented here will enhance our understanding of the evolutionary relationships within Sturnidae *sensu lato*.

## Figures and Tables

**Figure 1 animals-14-02777-f001:**
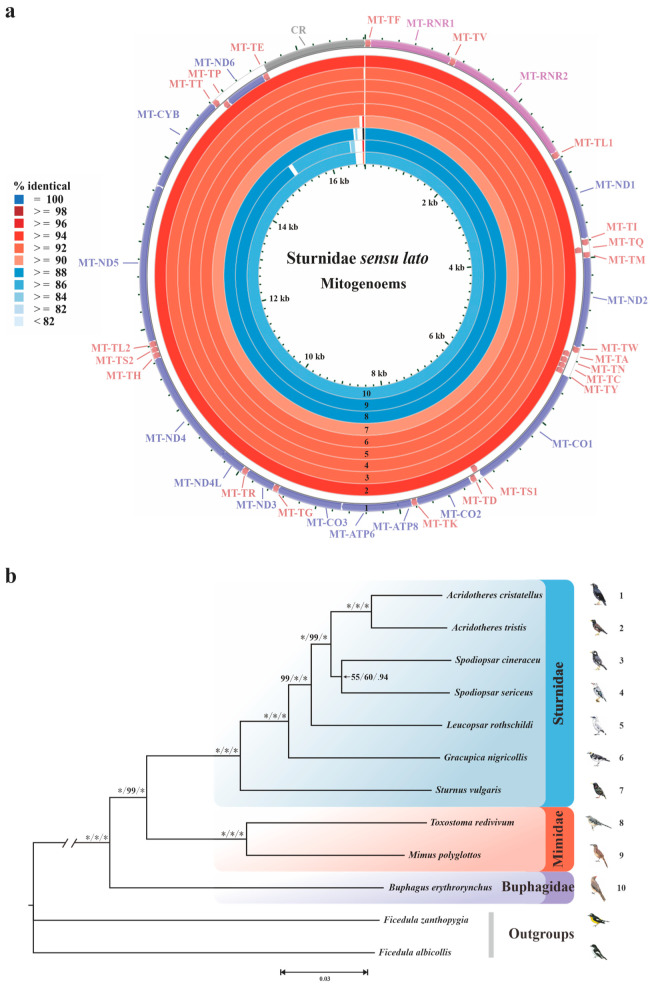
(**a**) Circular map of mitogenomes of Sturnidae *sensu lato*. The different colors represent the BLAST-identical percentages. The mitogenomes from outside to inside are as follows (labeled 1 to 10, respectively): *A. cristatellus*, *A. tristis*, *S. cineraceus*, *S. sericeus*, *L. rothschildi*, *G. nigricollis*, *S. vulgaris*, *T. redivivum*, *M. polyglottos*, and *B. erythrorynchus*. (**b**) Phylogenetic tree of the relationships among the 10 Sturnidae *sensu lato* species based on 13 mitochondrial PCGs, with two Muscicapidae outgroups. The support values of each node are indicated in the order of ML, MP, and BI inferences, and “*” indicates full support.

**Figure 2 animals-14-02777-f002:**
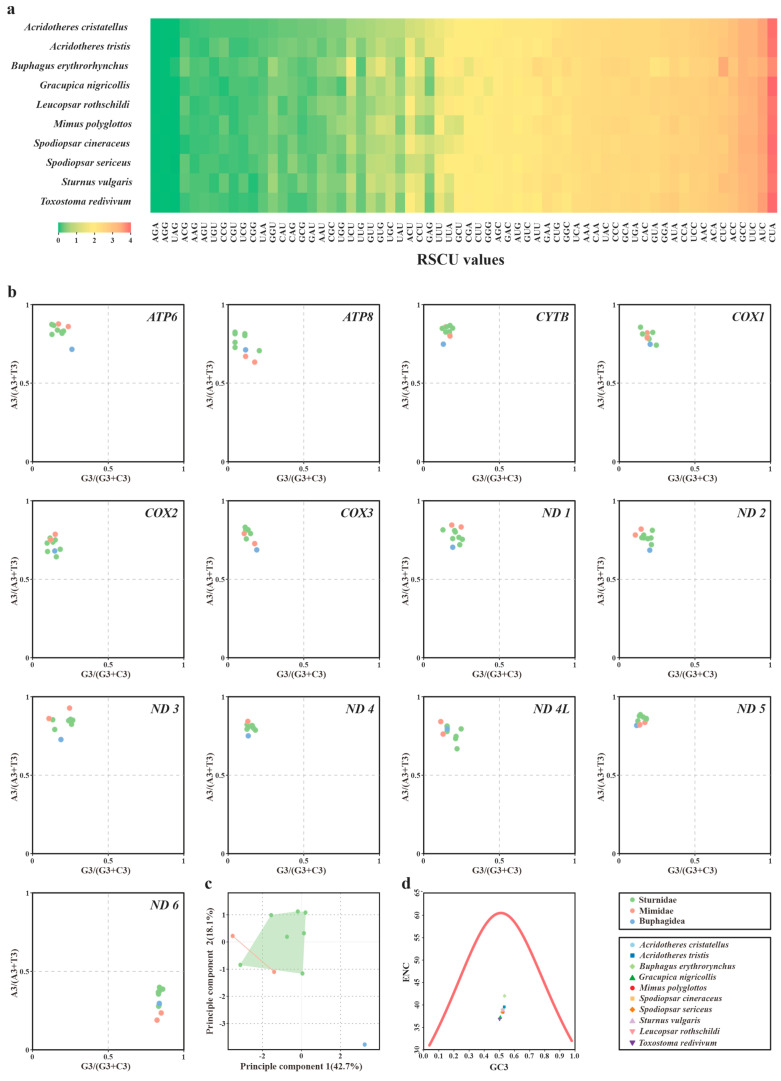
(**a**) RSCU heatmap of overall mitochondrial PCGs of Sturnidae *sensu lato*. (**b**) PR2 plots of each mitochondrial PCG of Sturnidae *sensu lato*. (**c**) PCA analysis based on the ENC values of each mitochondrial PCG. (**d**) The comparison of the ENC vs. GC3s curve of the PCGs in the mitogenomes of Sturnidae *sensu lato*. The continuous red line represents the expected ENC curve.

**Figure 3 animals-14-02777-f003:**
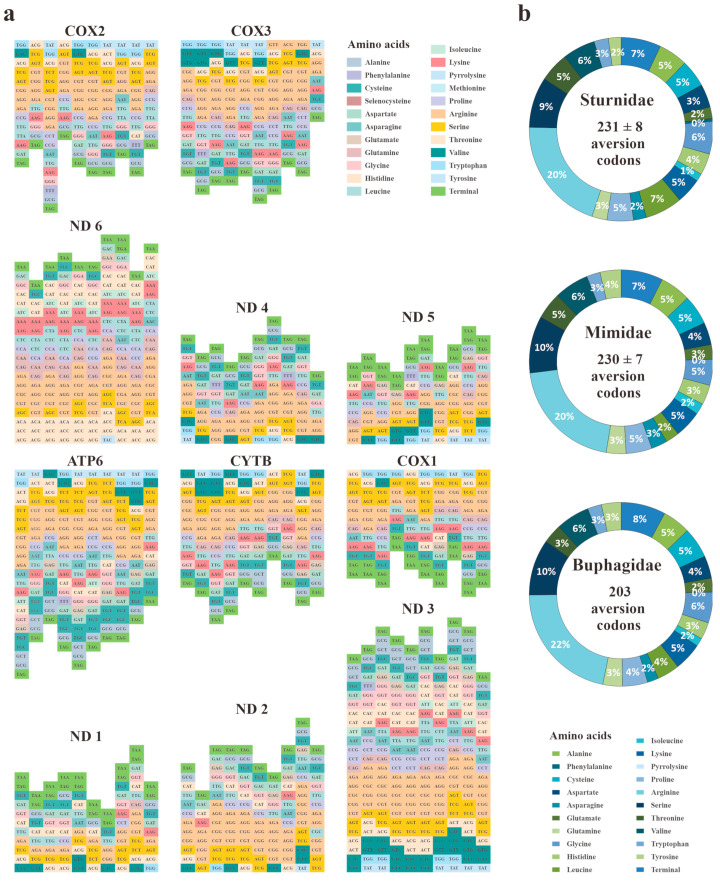
(**a**) Codon aversion motifs of ten investigated Sturnidae *sensu lato* mitochondrial PCGs. (**b**) Codon aversion numbers of the three families.

**Figure 4 animals-14-02777-f004:**
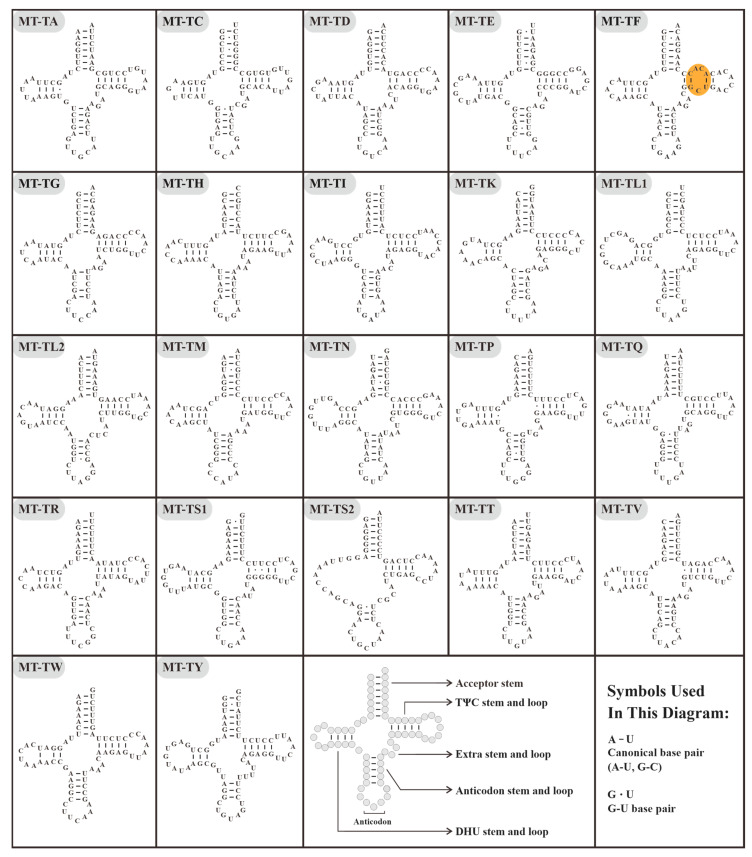
The presumed secondary structures of the tRNAs in *A. cristatellus*. The new form of the TΨC stem in MT-TF is displayed using an orange box.

**Figure 5 animals-14-02777-f005:**
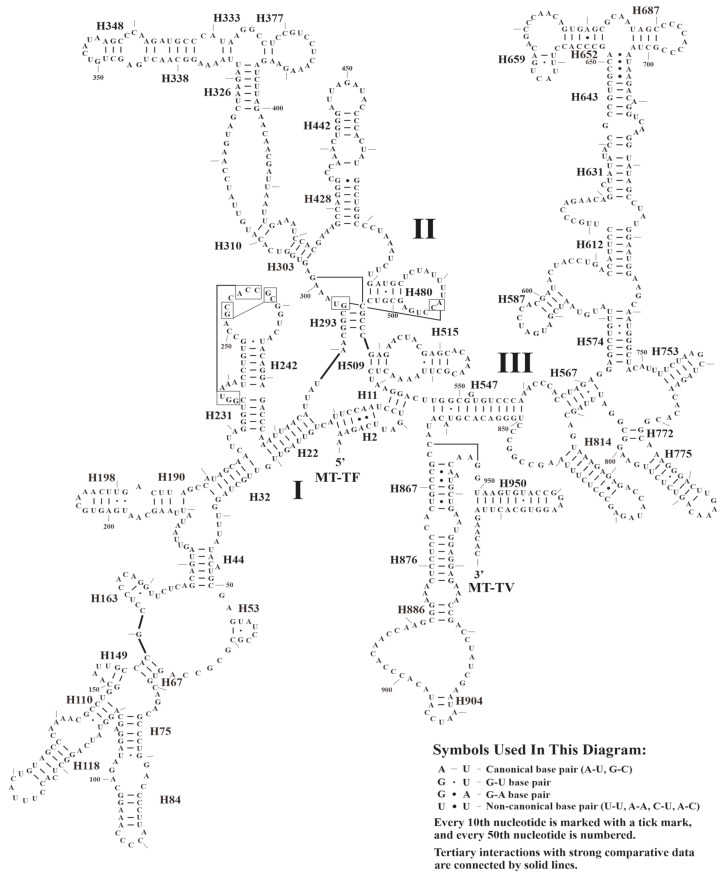
The predicted secondary structures of MT-RNR1 in *A. cristatellus*.

**Figure 6 animals-14-02777-f006:**
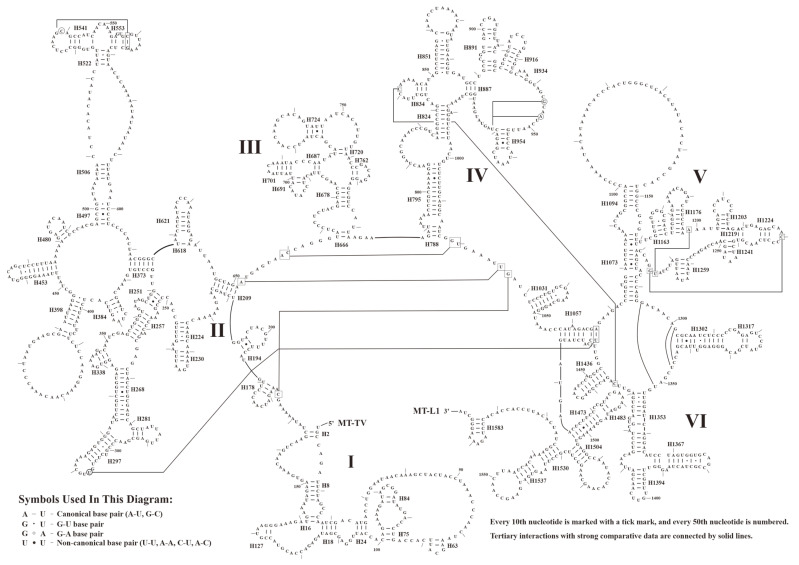
The predicted secondary structures of MT-RNR2 in *A. cristatellus*.

**Figure 7 animals-14-02777-f007:**
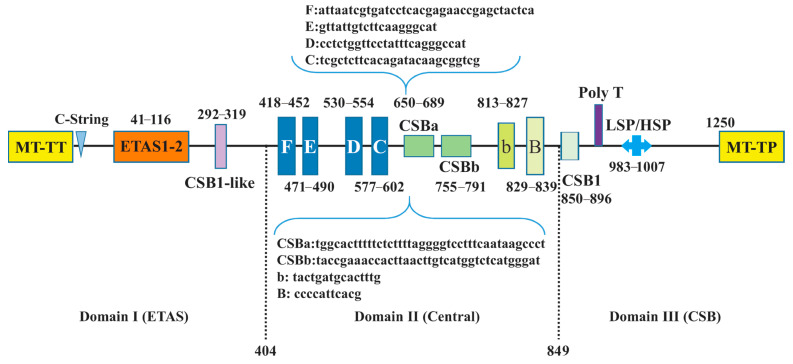
The structure of the control region in the mitogenome of *A. cristatellus*. Note that ETAS denotes extended termination-associated sequences, CSB denotes conserved sequence block, and HSP denotes heavy-strand transcription promoter.

**Figure 8 animals-14-02777-f008:**
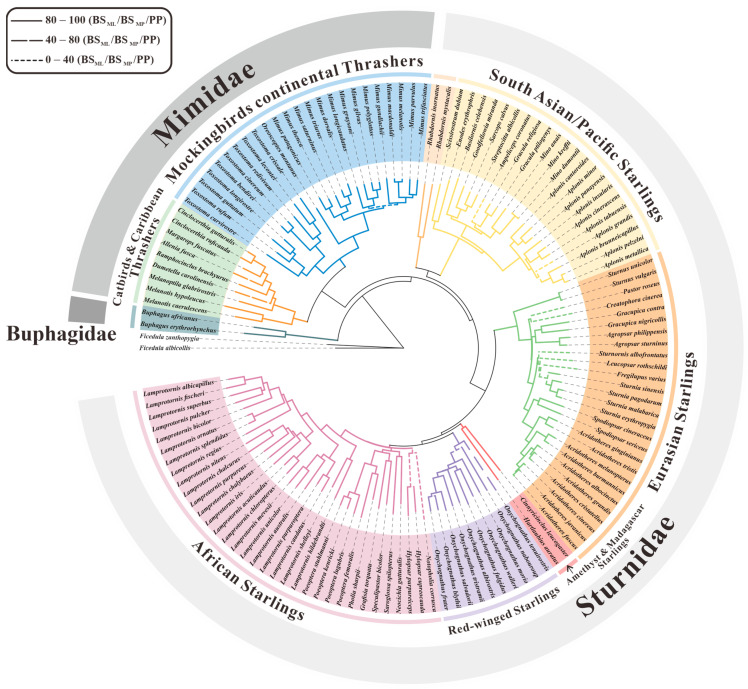
Nucleotide-based phylogenetic tree of 132 Sturnidae *sensu lato* taxa with two Muscicapidae outgroups. This analysis utilized a combined multilocus dataset (five mitochondrial genes and eight nuclear genes). The support value for each node is denoted using different line types.

**Table 1 animals-14-02777-t001:** The species of mitogenomes examined in this study.

Family	Species	Accession	Reference	Size (bp)/AT%					
				Heavy-Strand	PCGs	MT-RNR1	MT-RNR2	tRNA Genes	CR
Buphagidae	*Buphagus erythrorynchus*	MN356312	[56]	16,802/51.85	11,400/50.65	979/50.97	1602/54.24	1546/57.18	1231/54.51
Mimidae	*Mimus polyglottos*	BK010995 ^#^	This study	16,804/52.46	11,400/51.26	985/51.37	1600/54.44	1540/57.27	1231/56.38
Mimidae	*Toxostoma redivivum*	MN356247	[56]	16,780/52.91	11,400/52.13	983/50.76	1600/54.81	1542/56.49	1208/55.55
Sturnidae	*Acridotheres cristatellus*	NC_015613	This study	16,820/52.34	11,400/51.18	979/50.97	1598/54.76	1541/57.30	1250/55.52
Sturnidae	*Acridotheres tristis*	NC_015195	This study	16,822/51.99	11,400/50.87	979/50.97	1598/54.38	1542/57.00	1251/54.60
Sturnidae	*Gracupica nigricollis*	NC_020423	This study	16,839/52.95	11,400/51.96	978/50.92	1602/55.06	1541/58.08	1253/55.55
Sturnidae	*Leucopsar rothschildi*	MN356237	[56]	16,845/52.70	11,400/51.61	978/51.53	1600/55.13	1543/57.23	1254/55.58
Sturnidae	*Spodiopsar cineraceus*	NC_015237	This study	16,821/52.49	11,400/51.29	980/51.53	1598/55.19	1541/57.82	1249/54.68
Sturnidae	*Spodiopsar sericeus*	NC_014455	[57]	16,823/52.67	11,400/51.51	979/51.69	1598/55.07	1543/57.23	1250/55.76
Sturnidae	*Sturnus vulgaris*	NC_029360	[58]	16,793/52.38	11,400/51.29	974/50.72	1599/54.28	1540/57.60	1233/55.31

^#^ Mitogenomes retrieved from public database by third party annotation (TPA).

**Table 2 animals-14-02777-t002:** Rates and patterns of evolution among mitochondrial PCGs and species of Sturnidae *sensu lato*.

Gene	Length (bp)	Var. Sites [%]	π	dN	dS	dN/dS	ts/tv
ATP6	684	224 (32.75)	0.11816	0.1127	5.4502	0.02068	10.2768
ATP8	168	58 (34.52)	0.13294	0.433	1.6895	0.25631	5.4548
CO1	1551	369 (23.79)	0.09015	0.0191	3.3076	0.00579	6.1603
CO2	684	196 (28.65)	0.10679	0.0855	3.9109	0.02185	6.6734
CO3	786	215 (27.35)	0.0955	0.0756	4.1272	0.01833	7.3630
CYB	1143	292 (25.55)	0.10192	0.0844	4.7612	0.01772	6.4004
ND1	978	330 (33.74)	0.13263	0.0848	4.9668	0.01708	9.2547
ND2	1041	392 (37.66)	0.13797	0.2108	4.1455	0.05085	8.5414
ND3	351	115 (32.76)	0.12023	0.1351	3.7409	0.03611	8.9541
ND4	1380	457 (33.12)	0.12308	0.1077	7.2908	0.01477	12.7131
ND4L	297	97 (32.66)	0.1138	0.0906	7.2069	0.01257	11.8799
ND5	1818	570 (31.35)	0.10785	0.1326	5.9385	0.02232	9.6700
MT-ND6	519	178 (34.30)	0.13697	0.2388	2.5386	0.09406	6.3762
Overall	11,400	3493 (30.64)	0.1135				

**Table 3 animals-14-02777-t003:** The best Bayesian evolutionary models in the mitogenomic dataset.

Gene	Model	Model Setting	
		Lset nst	Rate
MT-ATP6	HKY+I+G4	2	invgamma
MT-ATP8	GTR+I	6	propinv
MT-CO1	GTR+G4	6	gamma
MT-CO2	HKY+I+G4	2	invgamma
MT-CO3	HKY+G4	2	gamma
MT-CYB	HKY+I+G4	2	invgamma
MT-ND1	HKY+I+G4	2	invgamma
MT-ND2	HKY+I+G4	2	invgamma
MT-ND3	HKY+G4	2	gamma
MT-ND4	GTR+G4	6	gamma
MT-ND4L	HKY+G4	2	gamma
MT-ND5	HKY+I+G4	2	invgamma
MT-ND6	HKY+I+G4	2	invgamma

**Table 4 animals-14-02777-t004:** Determination of the substitution saturation of the PCGs in the mitogenomic dataset.

Gene	Codon Position	Iss ^a^	Iss.cSym ^b^	*p*	Iss.cAsym ^c^	*p*
ATP6	1st	0.0877	0.6828	<0.0001	0.5256	<0.0001
	2nd	0.0201	0.6828	<0.0001	0.5256	<0.0001
	3rd	0.486	0.6828	<0.0001	0.5256	0.1973
ATP8	1st	0.1931	0.8013	<0.0001	0.8228	<0.0001
	2nd	0.1114	0.8013	<0.0001	0.8228	<0.0001
	3rd	0.3077	0.8013	<0.0001	0.8228	<0.0001
CO1	1st	0.0202	0.7237	<0.0001	0.5571	<0.0001
	2nd	0.0026	0.7237	<0.0001	0.5571	<0.0001
	3rd	0.4206	0.7237	<0.0001	0.5571	<0.0001
CO2	1st	0.0478	0.6828	<0.0001	0.5256	<0.0001
	2nd	0.0139	0.6828	<0.0001	0.5256	<0.0001
	3rd	0.4466	0.6828	<0.0001	0.5256	0.0125
CO3	1st	0.0407	0.6873	<0.0001	0.526	<0.0001
	2nd	0.0236	0.6873	<0.0001	0.526	<0.0001
	3rd	0.444	0.6873	<0.0001	0.526	0.0049
CYB	1st	0.0609	0.7051	<0.0001	0.5386	<0.0001
	2nd	0.0088	0.7051	<0.0001	0.5386	<0.0001
	3rd	0.4611	0.7051	<0.0001	0.5386	0.0042
ND1	1st	0.0734	0.6969	<0.0001	0.5316	<0.0001
	2nd	0.0168	0.6969	<0.0001	0.5316	<0.0001
	3rd	0.4984	0.6969	<0.0001	0.5316	0.1917
ND2	1st	0.1217	0.7001	<0.0001	0.5341	<0.0001
	2nd	0.0524	0.7001	<0.0001	0.5341	<0.0001
	3rd	0.4747	0.7001	<0.0001	0.5341	0.0189
ND3	1st	0.1075	0.6886	<0.0001	0.5759	<0.0001
	2nd	0.0635	0.6886	<0.0001	0.5759	<0.0001
	3rd	0.4817	0.6886	0.0001	0.5759	0.0643
ND4	1st	0.0871	0.7161	<0.0001	0.5494	<0.0001
	2nd	0.03	0.7161	<0.0001	0.5494	<0.0001
	3rd	0.4936	0.7161	<0.0001	0.5494	0.011
ND4L	1st	0.0978	0.7016	<0.0001	0.6103	<0.0001
	2nd	0.0203	0.7016	<0.0001	0.6103	<0.0001
	3rd	0.4469	0.7016	<0.0001	0.6103	0.0005
ND5	1st	0.0802	0.7349	<0.0001	0.5684	<0.0001
	2nd	0.0374	0.7349	<0.0001	0.5684	<0.0001
	3rd	0.4608	0.7349	<0.0001	0.5684	<0.0001
ND6	1st	0.1664	0.6787	<0.0001	0.5345	<0.0001
	2nd	0.0689	0.6787	<0.0001	0.5345	<0.0001
	3rd	0.4371	0.6787	<0.0001	0.5345	0.0135

^a^ Iss: index of substitution saturation. ^b^ Iss.cSym: critical index of substitution saturation (assuming a symmetrical topology). ^c^ Iss.cAsym: critical index of substitution saturation (assuming an asymmetrical topology).

## Data Availability

The sequence data generated in this study are available in GenBank of the National Center for Biotechnology Information (NCBI) under the access numbers NC_014455, NC_015195, NC_015237, NC_0s15613, and NC_020423.

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
