# Peer review of "Sturnidae sensu lato Mitogenomics: Novel Insights into Codon Aversion, Selection, and Phylogeny"

_animals, 2024, doi:10.3390/ani14192777_

Round 1
Reviewer 1 Report
Comments and Suggestions for Authors
This study has very pertinent considerations and valuable implications for the group studied, with unpublished data of interest to the scientific community. There are a few problems with the manuscript and I believe that after a good revision, it will be ready for publication.

Moderate editing of English language required.
Author Response
Thank you for your useful comments and suggestions for our manuscript. We have submitted the revised manuscript to MDPI for English language editing, and it has been completed. The certificate for English language editing and point-by-point response are in the attachment.

Reviewer 2 Report
Comments and Suggestions for Authors
The work represents a valuable contribution and is well-organized. However, I believe the manuscript requires an in-depth review, particularly in the discussion section, which needs further elaboration in several parts. For example, the section on the phylogeny is particularly frustrating, as the authors did not fully engage in a deep discussion, which I would expect for a paper of this nature. I have provided below comments to help the authors revise and improve their work:
General Structure and Content:
Positive Aspects: The manuscript is well-organized, and the data presented could significantly contribute to the field.
Concerns: The discussion section needs substantial improvement. The authors should focus more on discussing their data, exploring the concordances and discordances found with existing literature.
Phylogeny Discussion:
The discussion on phylogeny is particularly weak. Authors should engage more deeply with the existing hypotheses on phylogenetic relationships, comparing and contrasting their findings with those hypotheses to evaluate the explanatory power of their proposed phylogeny. Additionally, the concordance among different datasets and optimization criteria employed (e.g., Maximum Likelihood, Maximum Parsimony, and Bayesian methods) should be better addressed.
Specific Comments:
Line 51: The term "basal phylogenetic position" is problematic. Oxpeckers are simply the sister group of the clade formed by starlings and mimids. The term "basal" is widely used but is jargon that does not adhere to the fundamentals of systematic phylogenetics. Consider rephrasing or referring to relevant literature (e.g., “Which side of the tree is more basal?” https://doi.org/10.1111/j.0307-6970.2004.00262.x).
Line 52: The phrase "described species" is unnecessary. The current diversity is estimated at 123 species. The number of described species might be larger due to synonymization over time.
Line 58-59: If considerable research efforts have been focused on this group, please provide references for the most comprehensive work.
Line 62: Avoid phrases like "To our knowledge." Instead, make your statements and provide references.
Line 73: Spell out the acronym NCBI (National Center for Biotechnology Information) for clarity.
Line 135-137: Provide a rationale for using three optimization criteria (Maximum Likelihood, Maximum Parsimony, and Bayesian) in your analysis. A brief explanation with references would be helpful for readers.
Line 168: Table 1 is hard to read, likely due to PDF conversion. Ensure clarity in the final submission.
Line 279: Revisit the use of the term "more basal" (see the earlier comment on Line 51).
Lines 283-285: This section repeats the methods. Consider revising for clarity.
Line 331-332: Provide references.
Lines 336-341: The information on no strong codon bias is based on only five species. Can this be extrapolated to a group of over 100 species? Consider merging this paragraph with the following one to improve clarity and flow.
Line 376-378: This paragraph is somewhat distracting. Authors should focus on discussing their data. Some of the content here could be moved to the introduction.
Line 380-384: Discussing phylogenetic relationships based on geography (e.g., Southern Asia) seems misplaced. Concentrate on your data and avoid unnecessary theoretical discussions.
Lines 389-406: The discussion on phylogeny needs significant revision. It mostly repeats the results without sufficient comparison with previous hypotheses. Authors should engage more deeply with the existing literature and discuss the concordances and discordances found among different datasets and optimization criteria.
Comments on the Quality of English LanguageThe English in the manuscript requires an in-depth copyediting effort, especially after the recent revisions. I also believe the authors should clearly indicate that the English usage has been improved when submitting the revised version.
Author Response

(The authors gave the same response as above.)

Reviewer 3 Report
Comments and Suggestions for Authors
This manuscript describes five new complete mitochondrial genomes of a group of birds, and compares these with a larger set of mitogenomes. The methods are up-to-date and the results are described in sufficient detail. The study includes some tests that have rarely been applied to mitogenomes of birds and describes their relevancy for molecular biology and the study of evolution. The study suits the journal Animals well, and I recommend it for publication.
I only have a series of minor comments, suggestions and edits.
Line 19: between ‘for’ and ‘their’ insert the word ‘elucidating’
Line 142: the figure shows that Ficedula zanthopyga, not F. hypoleuca, was used
Methods: please describe how the substitution saturation tests were carried out. What does a significant value mean?
Table 1: please note that the three ‘unpublished’ sequences were actually published by Feng et al. (2020).
Feng, S., Stiller, J., Deng, Y., Armstrong, J., Fang, Q., Reeve, A. H., Xie, D., Chen, G., Guo, C., Faircloth, B. C., Petersen, B., Wang, Z., Zhou, Q., Diekhans, M., Chen, W., Andreu-Sánchez, S., Margaryan, A., Howard, J. T., Parent, C., Pacheco, G., Sinding, M. H. S., Puetz, L., Cavill, E., Ribeiro, A. M., Eckhart, L., Fjeldså, J., Hosner, P. A., Brumfield, R. T., Christidis, L., Bertelsen, M. F., Sicheritz-Ponten, T., Tietze, D. T., Robertson, B. C., Song, G., Borgia, G., Claramunt, S., Lovette, I. J., Cowen, S. J., Njoroge, P., Dumbacher, J. P., Ryder, O. A., Fuchs, J., Bunce, M., Burt, D. W., Cracraft, J., Meng, G., Hackett, S. J., Ryan, P. G., Jønsson, K. A., Jamieson, I. G., da Fonseca, R. R., Braun, E. L., Houde, P., Mirarab, A., Suh, A., Hansson, B., Ponnikas, S., Sigeman, H., Stervander, M., Frandsen, P. B., van der Zwan, H., van der Sluis, R., Visser, C., Balakrishnan, C. N., Clark, A. G., Fitzpatrick, J. W., Bowman, R., Chen, N., Cloutier, A., Sackton, T. B., Edwards, S. V., Foote, D. J., Shakya, S. B., Sheldon, F. H., Vignal, A., Soares, A. E. R., Shapiro, B., González-Solís, J., Ferrer-Obiol, J., Rozas, J., Riutort, M., Tigano, A, Friesen, V., Dalén, L., Urrutia, A. O., Székely, T., Liu, Y., Campana, M. G., Corvelo, A., Fleischer, R. C., Rutherford, K. M., Gemmell, N. J., Dussex, N., Mouritsen, H., Thiele, N., Delmore, K., Liedvogel, M., Franke, A., Hoeppner, M. P., Krone, O., Fudickar, A. M., Milá, B., Ketterson, E. D., Fidler, A. E., Friis, G., Parody-Merino, Á. M., Battley, P. F., Cox, M. P., Barroso Lima, N. C., Prosdocimi, F., Parchman, T. L., Schlinger, B. A., Loiselle, B. A., Blake, J. G., Lim, H. C., Day, L. B., Fuxjager, M. J., Baldwin, M. W., Braun, M. J., Wirthlin, M., Dikow, R. B., Ryder, T. B., Camenisch, G., Keller, L. F., DaCosta, J. M., Hauber, M. E., Louder, M. I. M., Witt, C. C., McGuire, J. A., Mudge, J., Megna, L. C., Carling, M. D., Wang, B., Taylor, S. A., Del-Rio, G., Aleixo, A., Ribeiro Vasconcelos, A. T., Mello, C. V., Weir, J. T., Haussler, D., Li, Q., Yang, H., Wang, J., Lei, F., Rahbek, C., Gilbert, M. T. P., Graves, G.R., Jarvis, E. D., Paten, B. & Zhang, G. 2020. Dense sampling of bird diversity increases power of comparative genomics. Nature 587: 252–257.
It would be good to mention somewhere in the paper that another mitogenome of Gracupica nigricollis (JQ003192) is actually a chimera with DNA of both Gracupica nigricollis and Acridotheres cristatellus (see Sangster & Luksenburg 2021 for details). It is good that this sequence was not used in the present study.
Sangster, G & Luksenburg, JA 2021. Sharp increase of problematic mitogenomes of birds: causes, effects and remedies. Genome Biology and Evolution 13(9), evab210.
Line 178: after ‘Codon usage bias’, insert (CUB)
Line 208: replace “varied completely with each other for” with “differed among”
Line 271: change “combined dataset” to “multi-locus dataset”
Table 4: Note that a P-value cannot be 0 (it would mean that it is impossible). Thus, replace all P-values of 0 with P<0.0001.
Table 4: what do the a, b and c substripts in the first row mean?
Line 277: replace “three-method trees” with “ML, MP and Bayesian trees”
Line 280: change “basal position with” to “sister-group relationship with”
Line 301: Pholia sharpie = Pholia sharpii
Line 301: “was currently treated as the only species of a monotypic genus” should be rephrased as “is currently treated as a monotypic genus”
Line 307: “of above six” = “of the six”
Line 309: a basal sister group = a sister group
Line 321: was found to harbore = comprised
Lines 338-339: I did not understand the phrase “which presented obvious considered biased”. Please rephrase.
Line 345: please omit “the” before natural selection
Line 386: basal sister to = sister to
Line 395: proved = demonstrate
Line 496: please italise the scientific names
Lines 505-506: please italise the scientific name
Lines 570-572: please note that the Sarkar paper has been retracted. The mitogenome was not of Turdoides affinis but represented a mitogenome of 5 different species!
Line 581: please italise the two genus names
Line 583: please italise the genus name
Line 585: please italise the genus name
Line 608: please italise the two genus names
Line 610: please italise the two genus names
Comments on the Quality of English LanguageI have suggested a couple of edits in my review.
Author Response
Thank you for your useful comments and suggestions for our manuscript. The point-by-point response is in the attachment.

Round 2
Reviewer 2 Report
Comments and Suggestions for Authors
Dear Authors,
I have completed my review of the revised version of your manuscript and enjoyed reading both the manuscript and your cover letter. I am pleased to know that you found my comments and suggestions helpful in revising and improving your text. I am satisfied with the changes you have made and see no further impediments to its acceptance.